# Uncovering the characteristics of the gut microbiota in patients with acute ischemic stroke and phlegm-heat syndrome

**Tingting Li**[1,2], **Qianhui Sun**[3], **Luda Feng**[1,2], **Dong Yan**[1,2], **Boyuan Wang**[1], **Mingxuan Li**[1,2], **Xuejiao Xiong**[1,2], **Dayong Ma**[1]*, **Ying Gao**[1,2]*

**1** Dongzhimen Hospital, Beijing University of Chinese Medicine, Beijing, China, **2** Institute for Brain Disorders, Beijing University of Chinese Medicine, Beijing, China, **3** Oncology Department, Guang'anmen Hospital, China Academy of Chinese Medical Sciences, Beijing, China

* ponymdy@126.com (DM); gaoying973@163.com (YG)

## Abstract

Growing evidence has indicated that the characteristics of gut microbiota are associated with acute ischemic stroke (AIS). Phlegm-heat syndrome (PHS), a specific pathological state of the AIS, is one of the common traditional Chinese syndromes of stroke. The long duration of PHS in patients with AIS could lead to poor clinical outcomes. Gut microbiota characteristics in patients with both AIS and PHS, and their relationship remains unknown. This study was designed to investigate the alterations in gut microbiota in patients with AIS and PHS through a cross-sectional study. Fecal samples were collected from 10 patients with AIS and non-PHS (ntAIS), 7 patients with AIS and PHS (tAIS), and 10 healthy controls (HC). Samples were profiled *via* Illumina sequencing of the 16S rRNA V3-V4. Stroke severity was assessed at admission by the National Institutes of Health Stroke Scale (NIHSS) and modified Rankin scale (mRS); their correlation with gut microbiota was investigated. The alpha-diversity of the bacterial communities was significantly higher in the fecal samples of patients with tAIS than in patients with ntAIS (Shannon index, P = 0.037). In addition, the combined tAIS and ntAIS group (tntAIS) exhibited higher microbiotic diversity when compared with HC (chao1, P = 0.019). The structure of intestinal microbiota was effectively distinguished between the tAIS and ntAIS group (ANOSIM, r = 0.337, P = 0.007). Additionally, the gut microbiota structure was significantly different between the tntAIS and HC groups (ANOSIM, r = 0.217, P = 0.005). The genera, *Ruminococcaceae_ UCG_002* and *Christensenellaceae_R-7_group*, were implicated in the discrimination of PHS from non-PHS. The order *Lactobacillales* and family *Lachnospiraceae* were significantly negatively correlated with NIHSS and mRS at admission (P < 0.05). By contrast, the order *Desulfovibrionales*, families *Christensenellaceae* and *Desulfovibrionaceae*, and genera *Ruminococcaceae UCG-014* and *Ruminococcaceae UCG-002* were significantly positively correlated with NIHSS and mRS at admission (P < 0.05). This study is the first to profile the characteristics of gut microbiota in patients with AIS and PHS, compared with those with non-PHS. The genera, *Ruminococcaceae_ UCG_002* and *Christensenellaceae_R-7_group*, may be objective indicators of this traditional Chinese medicine (TCM) syndrome in AIS. Furthermore, it provides a microbe-inspired biological basis for TCM syndrome differentiation.

---

**Data Availability Statement:** The data are all contained within the Supporting Information files.

**Funding:** This research was financially supported by National Key R&D program "Evidence-based

evaluation and mechanism of Chinese medicine intervention programs in the acute phase of stroke disease (Grant no.2018YFC1705000);the Dongzhimen hospital Beijing University of Chinese Medicine Project (Grant no. DZMKJCX-2020-003). The funder had no role in study design, data collection and analysis, decision to publish, or preparation of the manuscript.

**Competing interests:** The authors have declared that no competing interests exist.

**Abbreviations:** AIS, Acute Ischemic Stroke; PHS, Phlegm-Heat Syndrome; tAIS, Acute Ischemic Stroke and Phlegm-Heat Syndrome; ntAIS, Acute Ischemic Stroke and non-Phlegm-Heat Syndrome; HC, Healthy Controls; NIHSS, National Institutes of Health Stroke Scale; mRS, modified Rankin Scale; DSSEIS, Diagnostic Scale of Syndrome Elements in Ischemic Stroke; OTU, Operational Taxonomic Unit; PCoA, Principle Coordinates Analysis; LEfSE, Linear discriminant analysis Effect Size; FDR, false discovery rate; LDA, linear discriminant analysis.

# Introduction

Stroke is the second-leading cause of death and disability worldwide; one out of four individuals will experience stroke at least once in their lifetime [1]. Stroke could be categorized into two types: ischemic and hemorrhagic stroke. It was estimated that approximately 62.4% of all stroke events were ischemic in 2009 [2]. The typical clinical manifestation of AIS is the neurological deficit over a single cerebral arterial vascular territory [3]. Additionally, at least half of the patients experienced gastrointestinal complications after stroke, including intestinal motility dysfunction, intestinal flora disturbance, leaky gut, intestinal bleeding, and enteropathogenic sepsis [4].

Recent studies have indicated that the dysbiosis of gut microbiota plays an essential role in the pathophysiology of neurological diseases, such as Alzheimer's disease, multiple sclerosis (MS), Parkinson's disease (PD), and stroke [5]. Additionally, alterations in gut microbiota are a risk factor for stroke; elevated concentrations of opportunistic pathogens and a decreased abundance of butyrate-producing bacteria (BPB) may help to identify individuals who are at high- or low-risk for stroke [6].

Trimethylamine N-oxide (TMAO), a key gut metabolite, which has been shown to play an significant role in the onset, development, and progression of stroke [7]. Interestingly, TMAO has been reported to induce platelet hyperreactivity as a potential mechanism to increase thrombotic risk, suggesting that TMAO increases as a potential risk for acute ischemic events [8]. An animal study demonstrated that the short-chain fatty acid (SCFA) metabolized by gut bacteria could contribute to poststroke neuronal plasticity through promoting microbiota activation and behavioral recovery measured by motor deficits of the affected forelimb [9], indicating that intestinal microbiota modulation could be a possible therapeutic target for the recovery of stroke. Moreover, studies have confirmed the existence of a bidirectional microbiota–gut–brain axis and the potential of microbiota-based inventions to improve stroke outcomes [10].

Guided by the theories of Chinese medicine, syndrome differentiation remains the core of treatment. A syndrome is commonly known as 'zheng hou' in Chinese, which is the awareness of the law of occurrence, development, and manifestation of disease [11]. Syndromes or patterns ("Zheng" in Chinese), defined as symptom clusters, are the specific intervention targets in Chinese medicine theory [12,13]. Phlegm-heat syndrome (PHS) consists of two syndrome elements, phlegm-damp and heat-flaming. It is the key syndrome in the acute phase of ischemic stroke. Prior studies have indicated that the longer duration of PHS is associated with more severe neurological deficits [14]. In cases of PHS stroke patients, the plasma motilin (MTL) level was obviously increased compared with the other syndrome types [15], which could further elucidate that stroke patients of PHS were more likely to suffer from the slowing of gastric emptying to release more MTL via neurosecretory feedback.

Though studies have extensively reported on the relationship between stroke and gut microbiota, little is known about the microbiotic features of the specific TCM syndrome in stroke. In this study, we aimed to explore the alterations of gut microbiota in patients with AIS and PHS and find the potential indicators to determine and differentiate the profile of PHS.

# Materials and methods

## Participants

This study was carried out in accordance with the Declaration of Helsinki and approved by the Ethics Committee of the Dongzhimen Hospital, Beijing University of Chinese Medicine.

Written informed consent was obtained from all patients or their legally authorized representatives prior to enrollment in the study.

Fecal samples were collected from patients with AIS and non-PHS (ntAIS group, n = 10); AIS and PHS (tAIS group, n = 7); and healthy controls (HC group, n = 10). Recruitment began in November 2021 and continued until the end of January 2022. Eligible subjects were male or female, from the age of 18–85 years. The baseline characteristics of the three groups were consistent and comparable, including age, sex, the body mass index, and comorbidities in order to exclude the influence of specific comorbidities (hypertension, diabetes, and hyperlipidemia) and other confounders that affect gut microbiota (Table 1; all P > 0.05). All patients with AIS within seven days of symptom onset were diagnosed according to the guidelines and confirmed by magnetic resonance imaging or computed tomography. The inclusion criterion of the ntAIS group was to have a phlegm-damp and heat-flaming syndrome element score of ≤4. A two syndrome element score of ≥10 was required for inclusion in the tAIS group. These scores were assessed according to Diagnostic Scale of Syndrome Elements in Ischemic Stroke (DSSEIS). Stroke patients with a history of intestinal diseases, intracerebral hemorrhage, other neurological disorders, psychiatric diseases, infectious diseases, autoimmune diseases, malignant tumor or exposure to antibiotics, probiotics, glucocorticoids, or immunosuppressants within 1 month before sample collection were excluded. Patients who were pregnant or reported alcohol abuse were also ineligible for this study. The HC group defined as volunteers absent from parenchymal lesions of the major organs and past medical history of cardio-cerebrovascular disease through physical examination. HC were screened and excluded if they had taken antibiotics, probiotics, glucocorticoids, or immunosuppressants within one month of the study start date. To exclude the effect of diet on the intestinal microflora, a medical doctor who was also a registered nutritionist in Dongzhimen hospital provided a standardized recipe designed for the paticipants enrolled in this study (S1 Table). Diets formulated in this recipe could also meet the daily energy requirements for healthy people. Three meals per day for all the participants in this study were prepared strictly according to the standardized recipe.

## Sample collection and DNA extraction

The first available fresh fecal samples were collected using standard sterile stool collection tubes. Fecal samples from the participants were collected within 48h after the participants were admitted to the hospital. All samples were transported in liquid nitrogen and stored at −80˚C. Bacterial genomic DNA was extracted using MN NucleoSpin 96 Soi Kits, according to the manufacturer's instructions.

**Table 1. Baseline characteristics of each group.**

|  | tAIS (n = 7) | ntAIS (n = 10) | HC (n = 10) | P value |
|---|---|---|---|---|
| Age (year) | 71.57 ± 9.95 | 67.50 ± 13.79 | 63.30 ± 7.29 | 0.310 |
| Male (%) | 4 (57.1) | 5 (50) | 5 (50) | 0.948 |
| BMI (kg m$^{-2}$) | 23.43 ± 1.16 | 23.98 ± 1.23 | 22.59 ± 1.57 | 0.137 |
| Hypertension (%) | 4 (57.1) | 6 (60) | 3 (30) | 0.348 |
| Diabetes (%) | 3 (42.9) | 5 (50) | 3 (30) | 0.655 |
| Hyperlipidemia (%) | 5 (57.1) | 4 (40) | 2 (20) | 0.105 |

Data expressed as n (%) or mean ± SD. AIS: Acute ischemic stroke, tAIS: Patients with PHS, ntAIS: Patients with non-PHS, HC: Healthy controls, BMI: Body mass index.

### PCR amplification and sequencing

To amplify the V3–V4 region of the 16S rRNA gene for Illumina deep sequencing, the universal primers, 338F: 5'-ACTCCTACGGGAGGCAGCA-3' and 806R: 5'-GGACTACHVGGGTW TCTAAT-3' were used. The PCR amplification was performed in duplicate in a total reaction volume of 20 μL: $H_2O$, 13.25 μL; 10× PCR ExTaq Buffer, 2.0 μL; DNA template (100 ng/mL), 0.5 μL; prime1 (10 mmol/L), 1.0 μL; prime2 (10 mmol/L), 1.0 μL; dNTP, 2.0 μL; and ExTaq (5 U/mL), 0.25 μL. After an initial denaturation at 95˚C for 5 min, amplification was performed, consisting of 30 cycles of incubations for 30 s at 95˚C, 20 s at 58˚C, and 6 s at 72˚C. This was followed by a final extension step at 72˚C for 7 min. Next, the amplified products were purified and recovered using 1.0% agarose gel electrophoresis. Finally, the library construction and sequencing steps were performed by Beijing Biomarker Technologies Co. Ltd.

## Bioinformatics analysis

Bioinformatic analysis was completed on the BMKCloud Bioinformatic Platform (Biomarker Technologies, Beijing, China). To obtain raw tags, paired-end reads were merged by Flash ver.1.2.11 software [16]. Next, the raw tags were filtered and clustered in the next steps. The merged tags were compared to the primers; tags with >6 mismatches were discarded using the FASTX-Toolkit [17]. Tags with an average quality score <20 in a 50 bp sliding window were truncated using Trimmomatic [18]. Tags shorter than 350 bp were removed. We identified possible chimeras by employing UCHIME [19], a tool included in Mothur ver.1.30 software [20]. The denoised sequences were clustered using USEARCH ver.10.0 software [21]. Tags with ≥97% similarity were regarded as an operational taxonomic unit (OTU). Taxonomy was assigned to all OTUs by searching against the Silva database [22]. Alpha diversity analysis of gut microbiota (Chao, ACE, Shannon, and Simpson indices) were calculated by Mothur ver. 1.30 software. Beta diversity analysis was used to compare the structure of gut microbiota. Principle Coordinates Analysis (PCoA) was performed with the aid of ggplot2 R package. Linear discriminant analysis Effect Size (LEfSe) analysis was performed with the aid of huttenhower.sph.harvard.edu server to determine significant differences of bacteria among groups. The Kruskal-Wallis sum-rank test (a = 0.05) was firstly used to detect taxa that showed significantly different abundances. The false discovery rate (FDR) was the corrected P-value; FDR<0.05 was considered significant. Biological consistency between subclasses was then investigated by Wilcoxon rank-sum test. Finally, linear discriminant analysis (LDA) was used to estimate effect sizes for different levels of gut microbiota. The threshold on logarithmic score (LDA) for discriminative features was set to 4.0.

## Statistical analyses

Statistical processing was performed using the IBM Statistics SPSS ver.20.0 software. The data are presented as the mean value ± standard deviation (SD). *t*-tests were performed when data were normally distributed. Chi-square tests were used to examine the general characteristics of the study population. Intergroup differences in the abundance of intestinal flora were explored using the Wilcoxon rank-sum test. The association between intestinal flora and stroke severity at admission was determined using Spearman's rank correlation coefficient. The threshold for statistical significance was set to $p < 0.05$.

# Results

## Operational taxonomic unit distributions

We obtained 437 OTUs based on the 16S rRNA gene sequencing analysis (S2 Table). The tAIS, ntAIS, and HC groups had 416, 426, and 423 OTUs, respectively. Three and seven OTUs

were specific to the ntAIS and HC groups, respectively. There were 401 OTUs shared by tAIS, ntAIS, and HC groups, as shown on the Venn diagram (S3 Table) (Fig 1).

## Alpha and beta diversities

For the alpha-diversity (S4 Table), we found that there was no difference in Ace, Chao1, or the Simpson Index between the tAIS and ntAIS groups. The Shannon index of the tAIS group differed significantly from that of the ntAIS group, as described in Table 2. Next, the tAIS and ntAIS group were combined into a single tntAIS group; Chao1 of the tntAIS group differed significantly from that of the HC group, as shown in Table 3. Principle Coordinates Analysis (PCoA) indicated that the microbiota structure of tAIS was significantly different from that of the ntAIS group; therefore, analysis of similarities (ANOSIM) was performed. This indicated that the gut microbiota structure between the two groups differed significantly (ANOSIM, r = 0.337, P = 0.007) (Fig 2). Additionally, PCoA demonstrated that the microbiota structure of the combined tntAIS group was significantly different from that of the HC group, indicating that these two groups could be separated (ANOSIM, r = 0.217, P = 0.005) (Fig 3).

The maximum abundance of the gut microbiota in HC, ntAIS, and tAIS groups was assessed according to species abundance at the phylum level (S5 Table). The top ten of these phyla were ranked (Fig 4). The relative abundance of Firmicutes and Bacteroidetes in the ntAIS and tAIS group showed a trend of lower values than that of the HC group. By contrast, the relative abundance of Actinobacteria and Proteobacteria in the ntAIS and tAIS group showed a trend of higher values than that of the HC group. However, they failed to reach the level of statistical significance. The ntAIS and tAIS groups exhibited significantly higher levels of Verrucomicrobia and Synergistetes than the HC group (P < 0.05) (Fig 5).

At the genus level (S6 Table), the top 25 ranked highest abundance of gut microbiota in three groups is exhibited as a column diagram (Fig 6). The highest abundance of *Megamonas* was found in the HC group (P < 0.05), whereas this group had the lowest abundance of *Akkermansia* and *Olsenella* (P < 0.05). The abundance of *Veillonella* was highest in the ntAIS group (P < 0.05). *Megasphaera* and *Cloacibacillus* abundance was significantly higher in the ntAIS group (P < 0.05) when compared with the HC group, while the abundance of *Agathobacter* in the ntAIS group was lower than in the HC group (Fig 7) Additionally, Linear discriminant

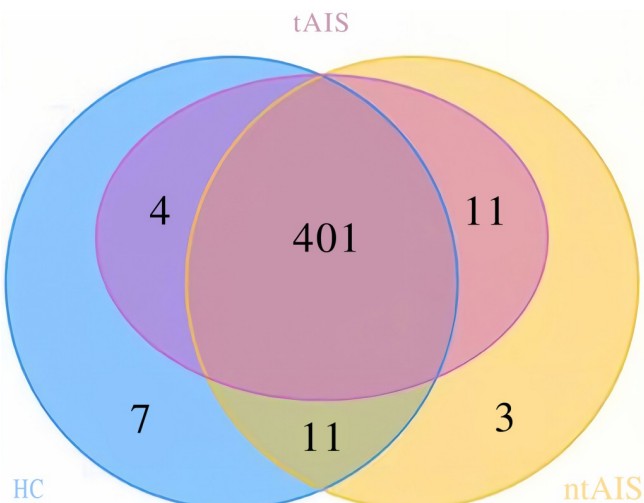

**Fig 1. Venn diagram showing OTU similarities and differences among three groups.**

**Table 2. Analysis of the α-diversity between the tAIS and ntAIS groups.**

| Parameter | tAIS (n = 7) | ntAIS (n = 10) | P value |
|---|---|---|---|
| Ace | 362.18 ± 8.35 | 342.97 ± 9.97 | 0.186 |
| Chao1 | 369.91 ± 9.94 | 347.18 ± 8.96 | 0.115 |
| Simpson | 0.93 ± 0.005 | 0.89 ± 0.02 | 0.059 |
| Shannon | 5.01 ± 0.11 | 4.43 ± 0.11 | 0.037* |

Data expressed as mean ± SD. AIS: Acute ischemic stroke, tAIS: Patients with PHS, ntAIS: Patients with non-PHS.

analysis Effect Size (LEfSE) analysis was evaluated using linear discriminant analysis (LDA) scores. These were calculated for the taxa of groups with significant differences in abundance to assess the influence of specific taxa between groups. At the genus level, Prevotella_9, Agathobacter, and Faecalibacterium were significantly more abundant in the HC group, while Escherichia_Shigella, Akkermansia, Bifidobacterium, and Ruminococcaceae_UCG_014 were significantly more enriched in the tntAIS group (Fig 8). Furthermore, the genera, Ruminococcaceae_UCG_002 and Christensenellaceae_R-7_*group*, were significantly more abundant in the tAIS group, the abundance of Megasphaera and Escherichia_Shigella was highest in the ntAIS group (Fig 9).

Finally, we assessed the relationship between stroke severity and gut microbiota using Spearman's rank correlation coefficient. Disease severity was assessed at admission using the NIHSS and mRS. At order level, Lactobacillales and *Desulfovibrionales* were significantly negatively and positively correlated with NIHSS and mRS scores, respectively (P<0.05) (Fig 10). At family level, *Christensenellaceae* and *Desulfovibrionaceae* were significantly positively correlated with NIHSS and mRS scores at admission (P < 0.05), while *Lachnospiraceae* was negatively correlated (P < 0.05) (Fig 11). At genus level, *Ruminococcaceae UCG-014* and *Ruminococcaceae UCG-002* were significantly positively correlated with NIHSS and scores (P < 0.05) (Fig 12).

## Discussion

Microbiota are found in all multicellular organisms, including in the human gut, where they mutualistically support host health regulation. Changes in the gut microbiota may be a risk factor and contribute to AIS. Furthermore, stroke can lead to microbiome dysbiosis. Interactions between intestinal microbiota and AIS with a certain TCM syndrome, such as PHS, is an emerging focus for research. The shannon index were higher in tAIS group than those in ntAIS group with statistically significant difference. Additionally, the chao1 index were significantly higher in the tntAIS group than those in HC group. Therefore, it could be concluded that AIS may result in the increased species richness of gut microbita, especially in AIS patients with PHS. The results of multiple diversities used to analyze the alpha and beta diversity of the

**Table 3. Analysis of the α-diversity between the tntAIS and HC groups.**

| Parameter | tntAIS (n = 17) | HC (n = 10) | P value |
|---|---|---|---|
| Ace | 350.88 ± 7.02 | 330.74 ± 8.49 | 0.085 |
| Chao1 | 356.54 ± 7.05 | 328.45 ± 8.39 | 0.019* |
| Simpson | 0.904 ± 0.001 | 0.896 ± 0.0096 | 0.662 |
| Shannon | 4.67±0.14 | 4.55±0.11 | 0.571 |

Data expressed as mean ± SD. AIS: Acute ischemic stroke, tntAIS: A combination of patients with PHS and non-PHS, HC: Healthy controls.

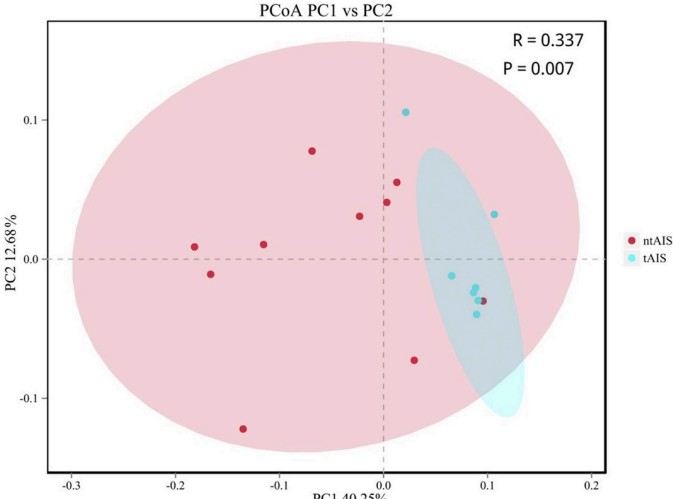

**Fig 2. PCoA analysis of the microbiota between tAIS and ntAIS groups.**

community indicated that the alterations in richness and structure of gut microbiota existed in patients with AIS and more prevalent in those patients who showed signs of PHS. Further- more, the genera, *Ruminococcaceae_ UCG_002* and *Christensenellaceae_R-7_group*, were implicated in the discrimination between PHS and non-PHS in patients with AIS. This could explain, at least in part, why tAIS patients are more likely to suffer from gastrointestinal com- plications and associated with more severe neurological deficits. Additionally, we found some gut microbiota were correlated with the stroke severity at admission.

In this study, the ntAIS and tAIS groups exhibited more abundant *Actinobacteria* and *Pro- teobacteria* than the HC group. A previous study found that the enzymes encoded by the CutC/D genes that were widely present in gut microbiota, including *Actinobacteria* and *Pro- teobacteria*, mediated the production of trimethylamine (TMA) from choline-rich food [23], indicating that patients with AIS harbored higher levels of TMA. Trimethylamine N-oxide (TMAO) is generated by the further transformation of flavin monooxygenases (FMO) acting

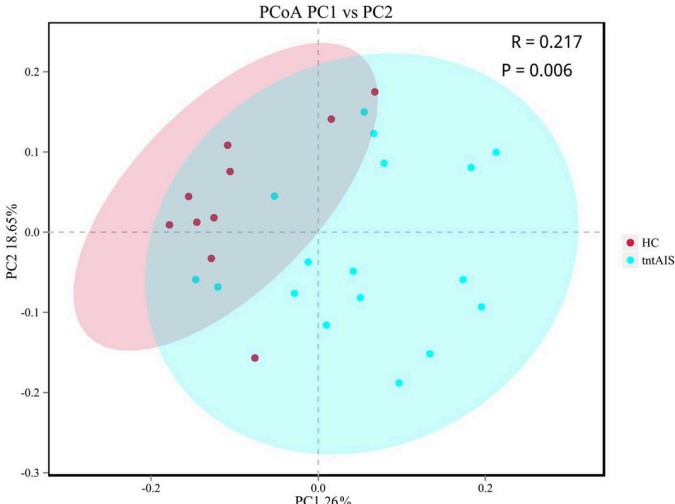

**Fig 3. PCoA analysis of the microbiota between HC and tntAIS groups.**

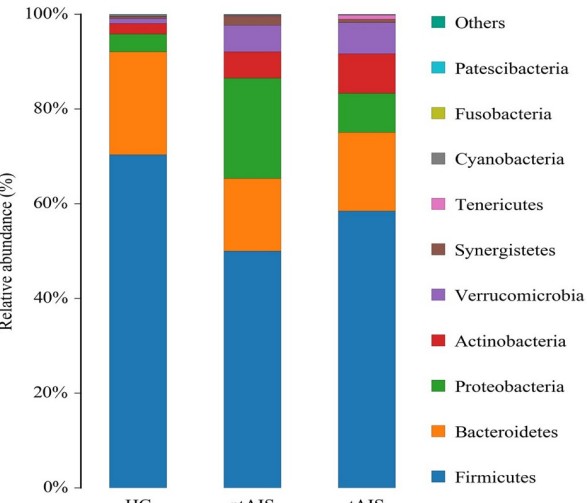

**Fig 4. The relative taxa abundance among the three groups at the phylum level.** AIS: Acute ischemic stroke; tAIS: Patients with PHS; ntAIS: Patients with non-PHS; HC: Healthy controls. The relative species abundances were calculated as percentages of the total species abundances(the number of tags measured in the sample divided by the total number of tags).

on TMA in the liver [24]. Several studies have reported that levels of TMAO in the circulation exacerbate the development of atherosclerosis [25,26]. Mouse studies have found that dietary supplementation of TMAO could enhance the atherosclerosis. Further, antibiotic treatment was used to block the production of TMA, which in turn reduced the atherosclerosis [26]. Moreover, it has been shown that high levels of TMAO are linked with high cardiovascular risk, including death, myocardial infarction and stroke [27]. Notably, it has been demonstrated that stroke led to activation of the sympathetic nervous system and triggered a chain of events that rapidly increased intestinal permeability and damaged the host's intestinal defenses [28], which resulted in further increase of TMAO in blood circulation.

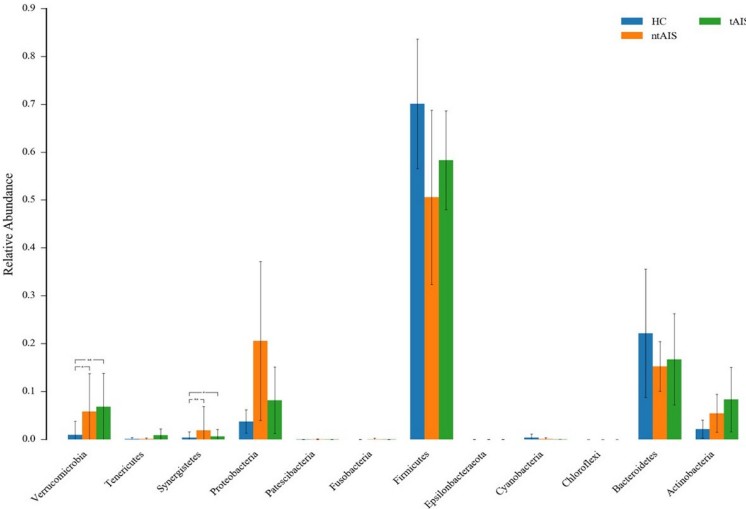

**Fig 5. The most abundant microbiotic components of three groups at the phylum level.** AIS: Acute ischemic stroke; tAIS: Patients with PHS; ntAIS: Patients with non-PHS; HC: Healthy controls.

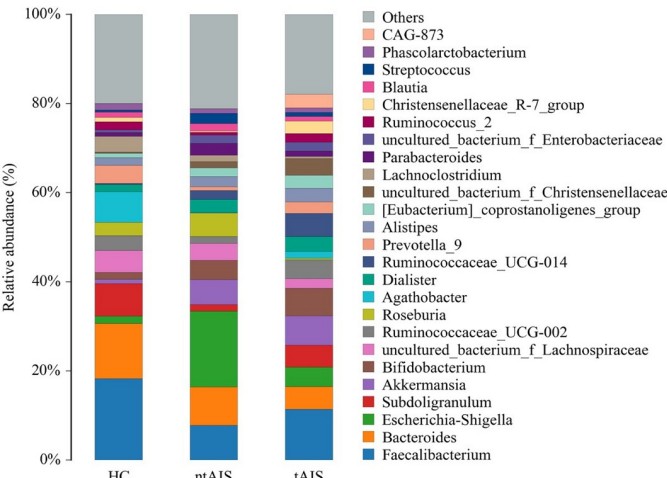

**Fig 6. The relative taxa abundance among the three groups at the genus level.** AIS: Acute ischemic stroke; tAIS: Patients with PHS; ntAIS: Patients with non-PHS; HC: Healthy controls. The relative species abundances were calculated as percentages of the total species abundances(the number of tags measured in the sample divided by the total number of tags).

In this study, the phyla, *Verrucomicrobia* and *Synergistetes*, were more abundant in both the ntAIS and tAIS groups than those of the HC group, which also showed the lowest abundance of *Akkermansia*. This eubacterium, easy to be determined in phylogenetic and metagenomic analyses [29,30], is the only cultured intestinal representative of *Verrucomicrobia*, and the increased abundance of *Akkermansia muciniphila* has been reported to be associated with the main alteration in mucosal microbiota composition caused by stroke, which is consistent with our results. The bacteria *Akkermansia muciniphila*, which can be found in the human intestine, is an actual species that uses mucin to produce short-chain fatty acid (SCFA) including

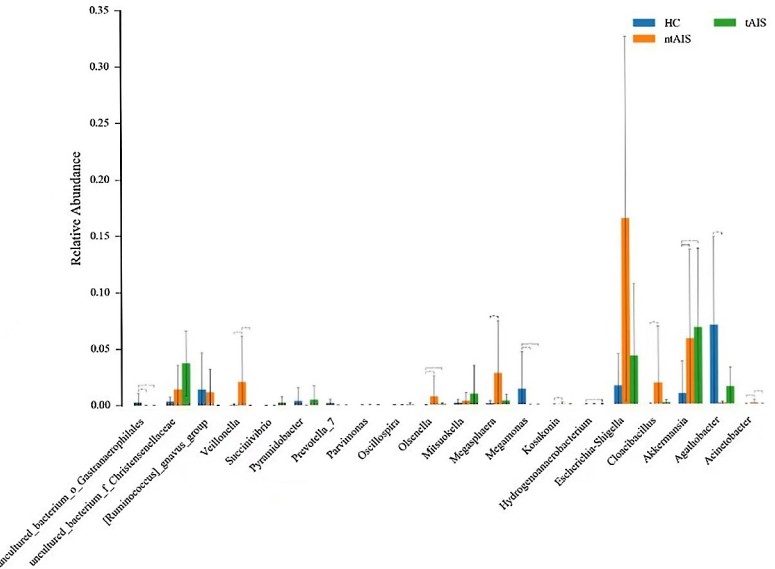

**Fig 7. The most abundant microbiotic components of three groups at the genus level.** AIS: Acute ischemic stroke; tAIS: Patients with PHS; ntAIS: Patients with non-PHS; HC: Healthy controls.

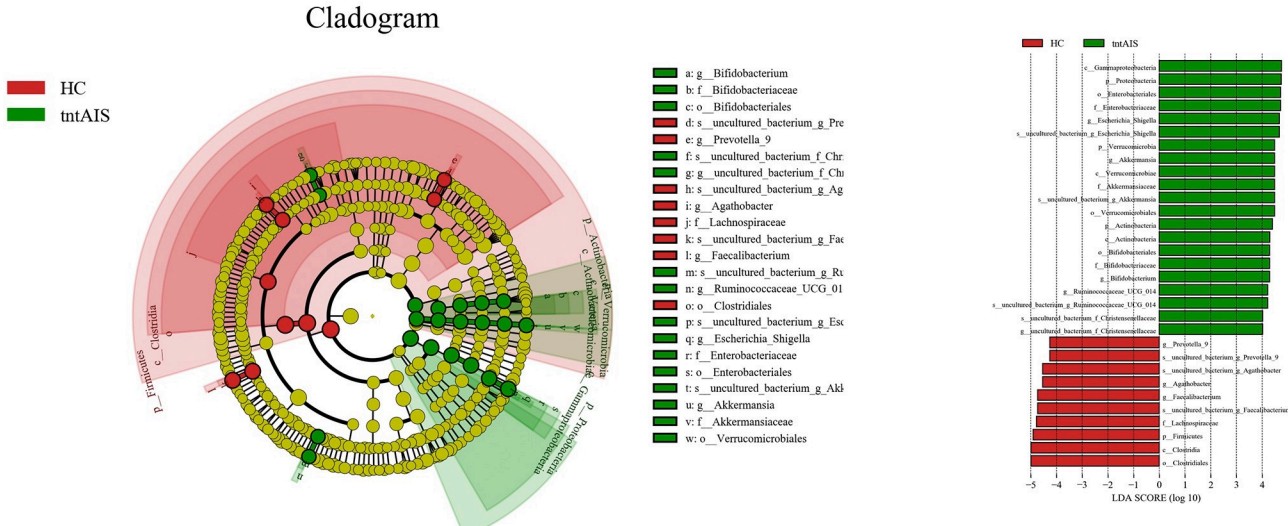

**Fig 8. Levels differ significantly between HC and tntAIS groups by LEfSE analysis.** AIS: Acute ischemic stroke, tAIS: Patients with PHS, ntAIS: Patients with non-PHS, tntAIS: A combination of patients with PHS and non-PHS, HC: Healthy controls. The LDA score was obtained by LDA (linear regression analysis); the larger the LDA score, the greater the influence of species abundance on the difference effect.

propionate and acetate. It was reported that high-level acetate could be consumed by butyrate-producing *Ruminococcaceae* to enhance the butyrate production [6]. Butyrate is an energy source that epithelial cells prefer, and which could help to maintain epithelial health [31]. Therefore, the *Akkermansia* may reinforce epithelial integrity through promoting the butyrate levels. Additionally, a mouse study indicated that traumatic brain injury (TBI) can lead to a disruption in mucosal barrier function and intestinal structure due to the failure in intestinal epithelial cell tight junctions [32]. Mucosal *Akkermansia* contributes to epithelial integrity and

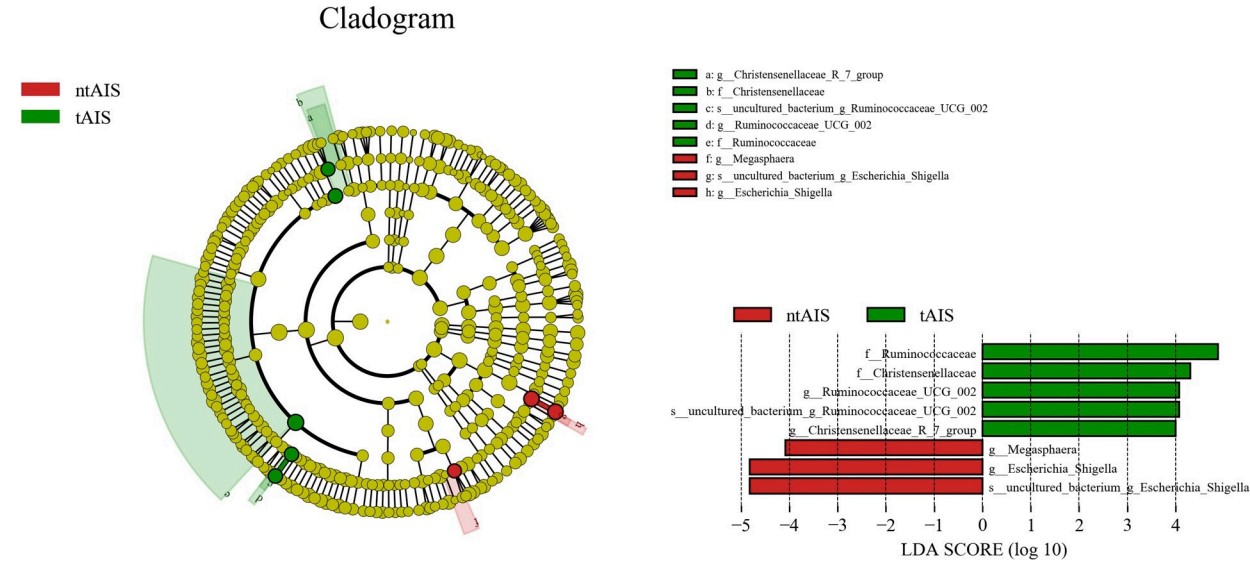

**Fig 9. Levels differ significantly between ntAIS and tAIS groups by LEfSE analysis.** AIS: Acute ischemic stroke, tAIS: Patients with PHS, ntAIS: Patients with non-PHS, tntAIS: A combination of patients with PHS and non-PHS, HC: Healthy controls. The LDA score was obtained by LDA (linear regression analysis); the larger the LDA score, the greater the influence of species abundance on the difference effect.

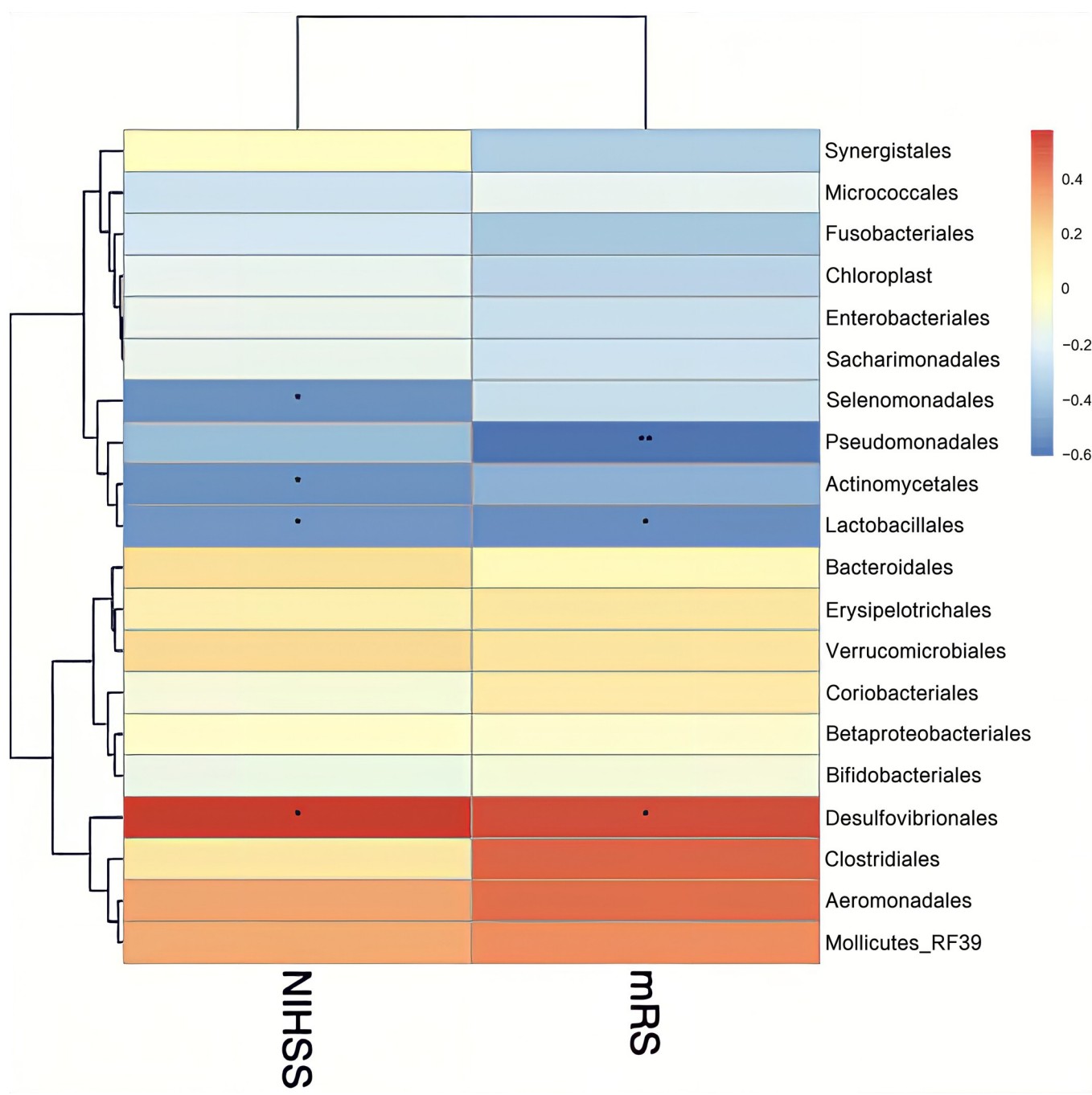

**Fig 10. Heatmap of the Spearman correlation coefficients between the gut microbiota at the order level and severity of stroke measured by NIHSS and mRS scores at admission.** *P < 0.05, **P < 0.01. NIHSS: National Institutes of Health Stroke Scale, mRS: modified Rankin Scale.

intestinal barrier fortification, which may account for the increased abundance of *Akkermansia* in ntAIS and tAIS groups.

The abundance of the pathogenic bacteria, including *Veillonella* and *Megasphaera*, was higher in the ntAIS group than in the HC group. Both bacteria are part of the family, *Veillonellaceae*. The increase in *Veillonellaceae* is associated with the elevated levels of circulating succinate [33], which is a microbe-metabolite detected in hypertension [34], type 2 diabetes

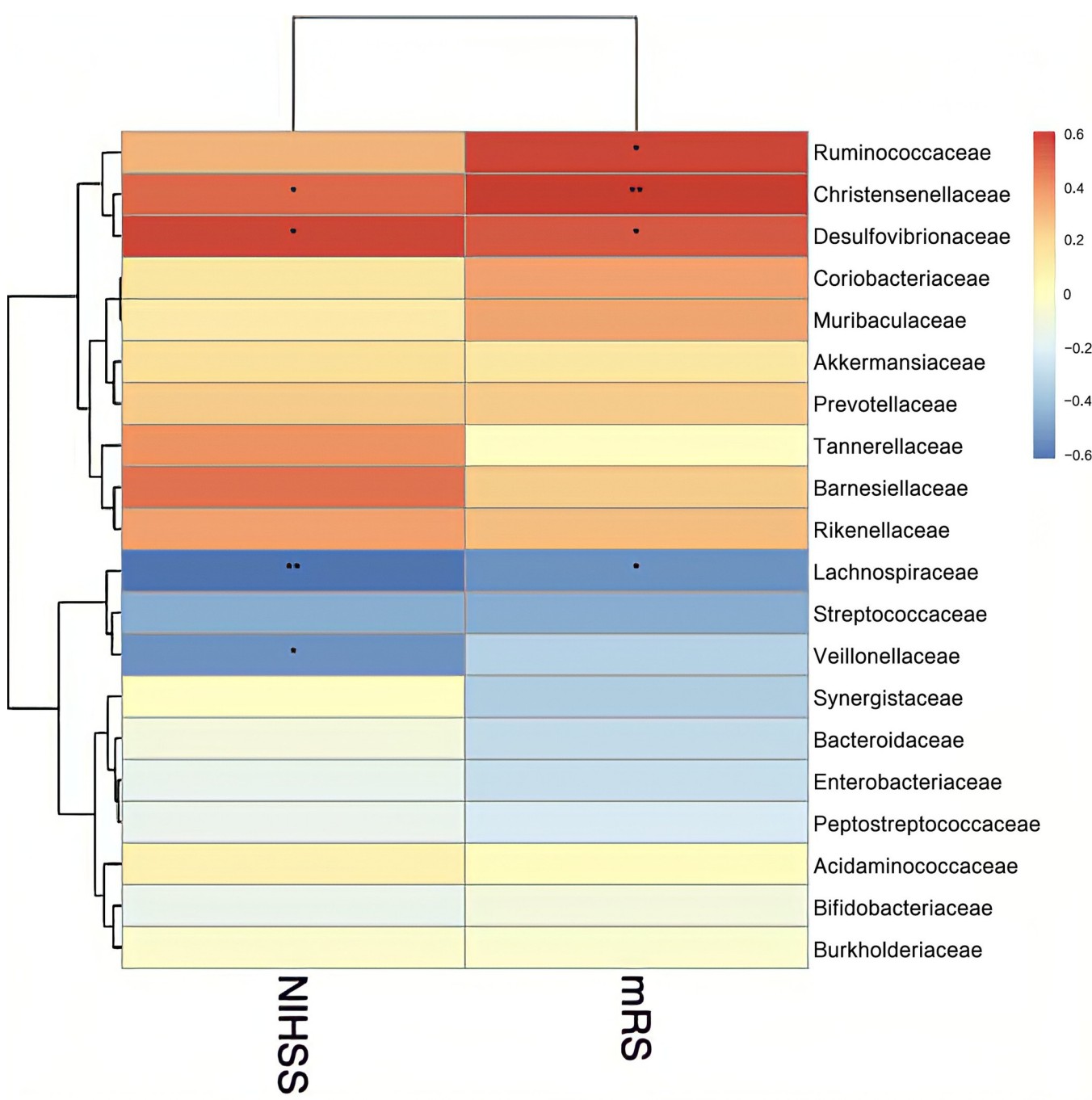

**Fig 11. Heatmap of the Spearman correlation coefficients between the gut microbiota at the family level and severity of stroke measured by NIHSS and mRS scores at admission.** *P < 0.05, **P < 0.01. NIHSS: National Institutes of Health Stroke Scale, mRS: Modified Rankin Scale.

mellitus (T2DM) [35], and obesity [36], which are the major *risk* factors of stroke as they predispose to atherosclerosis.

LEfSE analysis revealed that the genera, *Ruminococcaceae_UCG_002* and *Christensenellaceae_R-7_group* belonging to the class *Clostridia* in the phylum *Firmicutes* could assist in distinguishing between individuals in the tAIS and ntAIS groups because they were significantly more abundant in tAIS group. A previous study has demonstrated a positive correlation

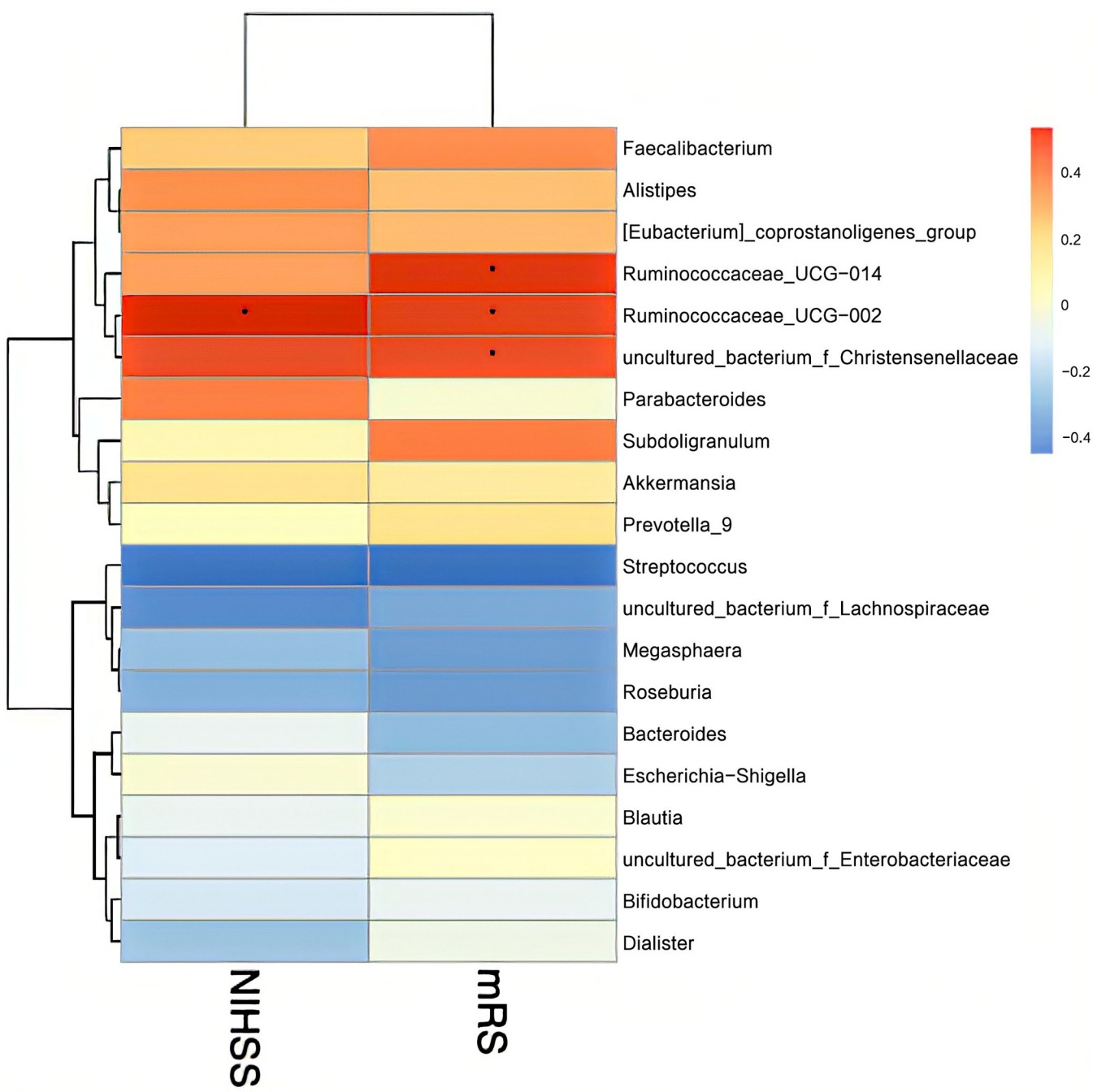

**Fig 12. Heatmap of the Spearman correlation coefficients between the gut microbiota at the genus level and severity of stroke measured by NIHSS and mRS scores at admission.** *P < 0.05, **P < 0.01. NIHSS: National Institutes of Health Stroke Scale, mRS: modified Rankin Scale.

between *Christensenellaceae_R-7_group* and NIHSS score in stroke patients [37], which is in line with our findings. To the best of our knowledge, our study is the first one to show that these gut microbiota can differentiate between tAIS and ntAIS groups. Our previous research found that patients with tAIS were more likely to suffer from constipation, more severe neurological deficits, and worse functional outcomes. Furthermore, a longer PHS duration was significantly associated with worse neurological deficits [14]. Therefore, early recognition,

prevention, and intervention of patients with AIS who have PHS is of significant implication. Several studies have reported that an increased abundance of *Christensenellaceae* is associated with the increased intestinal transit time, which could lead to constipation [38,39]. Previous studies have also demonstrated the relationship between an increased abundance of *Christensenellaceae* and other neurological disorders, including PD and MS [40,41]; nearly two-thirds of patients with these diseases suffered from intestinal motility disorders, especially constipation. Thus, the alterations in *Christensenellaceae_R-7_group* may contribute to the clinical features in patients with tAIS and early differentiation of this syndrome. However, future detailed research into the mechanisms of *Christensenellaceae_R-7_group* is required.

A male piglet study has shown a negative correlation between fecal *Ruminococcaceae* and brain N-acetylaspartate (NAA) [42]. The energy metabolizability of the acute phase of ischemic stroke was shown by the time-dependent decay of NAA, as measured by proton magnetic resonance spectra ($^1$H-MRS) [43,44]. The decrease in NAA was interpreted as irreversible neuronal loss and cellular metabolism impairment. Notably, the LEfSE analysis showed that the genus, *Ruminococcaceae_UCG_002* was more abundant in the tAIS group than in the ntAIS. Further, the genus, *Ruminococcaceae_UCG_014*, was more abundant in the combined tntAIS group than in the HC group. Spearman correlation coefficient analysis revealed a positive correlation between *Ruminococcaceae UCG-014*, *Ruminococcaceae UCG-002*, and NIHSS score at admission. It could offer preliminary evidence to some degree for the association between gut microbiota, traditional Chinese syndromes, and neuronal loss, which was further reflected by the fact that the tAIS group had more severe neurological deficits.

An increasing number of studies have focused on the short-chain fatty acid (SCFA) metabolites of gut bacteria, especially the butyric acid. The enzyme system encoded by the intestinal flora contains all known carbohydrate active enzymes (CAZYmes) and many potential novel CAZYmes. These can produce SCFAs through fermented dietary fiber and carbohydrates existing in the diet and gut mucus. Moreover, there are numerous physiological functions associated with SCFAs, including the regulation of immune responses, maintenance of gut barrier integrity, suppression of histone deacetylases and anti-inflammatory effects [23]. A previous study has reported that patients with high-risk of cerebral infarction exhibited a decrease in the abundance of butyrate-producing bacteria (BPB) [6]. Conversely, supplementation with butyric acid could reduce the leaky gut in patients with ischemic stroke [45]. Spearman's correlation revealed that *Lachnospiraceae* (a BPB) was negatively correlated with NIHSS and mRS scores at admission. Sodium butyrate (NaB), generated by BPB, is a histone deacetylase inhibitor that can traverse the blood-brain barrier. The NaB could promote the post-ischemic neurological recovery, decrease infarct volume, and reduce the expression of pro-inflammatory factors [46–48]. Additionally, the endothelial damage in atherosclerosis can be rescued by butyrate through multiples mechanisms, including the decreased expression of NADPH oxidase 2 and reactive oxygen species [49].

This was the first study to explore the characteristics of the gut microbiota in AIS patients with PHS. Ruminococcaceae_UCG_002 and Christensenellaceae_R-7_group could be applied to determine and differentiate the profile of PHS, was a key finding of this study and provided novel insight for a reasonable explanation on the cause of constipation and more severe neurological dysfunction in the AIS patients with PHS, which suggested that early diagnosis and early intervention was of significant implication in clinical practice. Additionally, this study also suggested that supplementation of the diet with fiber, butyrate, or probiotics containing butyrate-producing bacteria could exert the neuroprotective effect on stroke recovery via promotion of consolidate epithelial integrity. However, the main limitation of this study is the small sample size. The comparison between few fecal tAIS providers and the ntAIS patients may result in a failure to detect some certain bacteria acting as potential microbial indicators

to differentiate the PHS. Furthermore, further in-depth research is still required to understand causal mechanisms.

## Conclusion

This study characterized the alterations in gut microbiota composition in patients with tAIS and ntAIS. We found that the genera, *Ruminococcaceae_ UCG_002* and *Christensenellaceae_R-7_group* could be used as indicators for discriminating PHS in AIS, as well as being potential therapeutic targets or underlying advantages of TCM treatment. Furthermore, the mechanism of TCM treatment for PHS in AIS will be the subject of our future study.

## Supporting information

**S1 Checklist. STROBE checklist.**
(PDF)

**S1 Table. Standardized recipe designed according to the Chinese Food Guide Pagoda for participants in this study.**
(PDF)

**S2 Table. The number of OTUs identified from all samples.**
(XLS)

**S3 Table. Venn analysis performed at the OTU level among the three groups.**
(XLS)

**S4 Table. The α-diversity identified from all samples.**
(XLS)

**S5 Table. The relative taxa abundance between the three groups at the phylum level.**
(XLS)

**S6 Table. The relative taxa abundance between the three groups at the genus level.**
(XLS)

**S1 Data.**
(ZIP)

## Author Contributions

**Conceptualization:** Tingting Li, Qianhui Sun, Luda Feng, Dayong Ma, Ying Gao.

**Data curation:** Dong Yan, Xuejiao Xiong.

**Formal analysis:** Tingting Li, Qianhui Sun, Luda Feng, Boyuan Wang, Mingxuan Li.

**Investigation:** Tingting Li, Boyuan Wang, Mingxuan Li, Ying Gao.

**Project administration:** Ying Gao.

**Software:** Tingting Li, Qianhui Sun.

**Supervision:** Dayong Ma, Ying Gao.

**Visualization:** Tingting Li, Qianhui Sun, Luda Feng.

**Writing – original draft:** Tingting Li, Qianhui Sun, Luda Feng.

**Writing – review & editing:** Tingting Li, Dayong Ma, Ying Gao.

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
