## [Decision Letter · Decision Letter 0]

23 Aug 2022

PONE-D-22-09974Uncovering the characteristics of the gut microbiota in patients with acute ischemic stroke and phlegm-heat syndromePLOS ONE

Dear Dr. Gao

Thank you for submitting your manuscript to PLOS ONE. After careful consideration, we feel that it has merit but does not fully meet PLOS ONE’s publication criteria as it currently stands. Therefore, we invite you to submit a revised version of the manuscript that addresses the points raised during the review process. The risk factors for the healthy controls have to properly accounted for. The inclusion and exclusion of various subgroup studies (tAIS, ntAIS etc.) have to been fully rationalized. What criteria were utilized to differentiate between severe or not severe cases?  The significance of the variations in microbial population in these diseased conditions need to be appropriately commented on.

Please submit your revised manuscript by Sep 28 2022 11:59PM. If you will need more time than this to complete your revisions, please reply to this message or contact the journal office at plosone@plos.org. Please include the following items when submitting your revised manuscript:A rebuttal letter that responds to each point raised by the academic editor and reviewer(s). You should upload this letter as a separate file labeled 'Response to Reviewers'.A marked-up copy of your manuscript that highlights changes made to the original version. You should upload this as a separate file labeled 'Revised Manuscript with Track Changes'.An unmarked version of your revised paper without tracked changes. You should upload this as a separate file labeled 'Manuscript'.

We look forward to receiving your revised manuscript.

Kind regards,

Vasu D. Appanna

Academic Editor

PLOS ONE

Journal Requirements:

The authors declare that the research was conducted in the absence of any commercial or financial relationships that could be construed as a potential conflict of interest.

3. Please remove your figures from within your manuscript file, leaving only the individual TIFF/EPS image files, uploaded separately.  These will be automatically included in the reviewers’ PDF.

Reviewers' comments:

Reviewer's Responses to Questions

**Comments to the Author**

1. Is the manuscript technically sound, and do the data support the conclusions?

Reviewer #1: Partly

Reviewer #2: Yes

2. Has the statistical analysis been performed appropriately and rigorously? 

Reviewer #1: Yes

Reviewer #2: Yes

3. Have the authors made all data underlying the findings in their manuscript fully available?

Reviewer #1: Yes

Reviewer #2: Yes

4. Is the manuscript presented in an intelligible fashion and written in standard English?

Reviewer #1: Yes

Reviewer #2: Yes

5. Review Comments to the Author

Reviewer #1: Line 37: Adding one line to explain the connection between AIS and PHS would be helpful.

Line 76: affected

Line 100: (controls description. Non AIS, Non PHS?

Line 114: The absence of cardiovascular risk factors in the healthy control group makes them a pretty confounding variable compared to the other two groups, whereby conclusions about differences between HC and either of the other groups has to really be taken with a lot of salt. Perhaps it would be prudent to further elucidate why the decision was made to exclude non-stroke patients, who had hypertension/diabetes/hyperlipidemia.

Line 120: Is the recipe information included in the supplementary information?

Line 175: Where it shows the difference between the combined tntAIS and Chao1, does this not simply negate the value of including PHS in the analysis. Furthermore again we are confronted with he issue of confounds between chao1 and tntais. This is a serious flaw in the study design that needs to be addressed in terms of the logic behind combining the tAIS and ntAIS as well as the decision to exclude the confounds in the HC group.

Line 234: The NIHSS is a likert scale, not a continuous variable so what were the cutoffs used to characterize severity? Or were they treated as a continuous variable? Please clarify!

Reviewer #2: This study observes the characteristics of the gut microbiome in subjects with acute ischemic stroke and phlegm-heat syndrome.

1. I have no issue with the methodology and experimental approach of this research. I believe the techniques used such as 16S rRNA sequencing and bioinformatics approach are solid and the data presented confirms the conclusion of the study.

I have one minor recommendation: if possible, mention if the PCR amplification was performed one, in duplicate or triplicate for each subject.

2. The statistical analysis of the data is adequate. However, in one instance in the text, it is mention that the bacteria in the microbiome 'differed, but did not attain significance'. I think this needs to be rectified. If the statistic test doesn't show a significant difference, the text should not state that there is still a difference as this could lead to confusion.

Recommendation: adjust the text at lines 194-195 to address this.

3. I have no concern about data availability. There are supplemental files as well as supporting information available

4. The manuscript is written in acceptable English. There is no noticeable grammar/syntax/spelling mistakes. However, there are a few typos.

Line 276: typo in the reference citation

Line 297: typo in the reference citation

Other comments:

1) In the text body, Figure 3A is mentioned before Figure 2B. The text would flow better if figures were in order. If possible, please adjust the text or figure order. If not, kindly explain.

2) Figure 2: Mention in the legend how the abundance was calculated.

3) Figure 3: a) It would be pertinent to add the 'N' value for each group in the figure to avoid going back to the text. b) Describe 'relative abundance' in methods or in the legend. c) It is very hard to see what data is compared to what in the figure in the figure I was provided. I would recommend to adjust the figure to emphasize that better, or to improve the quality/resolution of the figure.

4) Figure 4: Mention what a low or high LDA score means.

5) Figure 2-3-4-5: I have some concerns about overall figure quality. Figures are blurry and hard to read. Overall quality or resolution of the figures needs to be improved, especially for the text found in the figure which is hard to read.

I recommend to go forward with the publication process once the points above are addressed.

6. PLOS authors have the option to publish the peer review history of their article (what does this mean?). If published, this will include your full peer review and any attached files.

Reviewer #1: **Yes: **Dr. Sean-Crispin Thomas

Reviewer #2: No

---

## [Author Response · Author response to Decision Letter 0]

29 Sep 2022

Dear Academic Editor Vasu D. Appanna,

Thank you for your letter and the opportunity to revise our paper on ‘Uncovering the characteristics of the gut microbiota in patients with acute ischemic stroke and phlegm-heat syndrome’ (Manuscript ID: PONE-D-22-09974). The suggestions offered by you and reviewers have been immensely helpful. We have addressed each concern or problem and describing the changes we have made. The revisions have been approved by all nine authors. We hope the revised manuscript will better suit PLoS One and are happy to consider further revisions. Also, we thank you for your continued interest in our research.

Sincerely,

Ying Gao on behalf of all co-authors

Q1.The risk factors for the healthy controls have to properly accounted for.

Author response: Thank you so much for your valuable advice. We were sorry that our unclear description made you puzzled by the definition of healthy control group. The HC group in this study was relatively healthy. Recent studies have indicated that the changes of intestinal flora were closely associated with diabetes, hypertension and hyperlipidemia. Intestinal microbiota disorder, such as the decrease of beneficial bacteria and the increase of harmful bacteria, is easy to occur in diabetes patients[1].Additionally, there are abnormal changes in the content of intestinal floras in patients with hypertension including the decreased quantities of Bifidobacterium and Bacteroides thetaiotaomicron and the increased number of Eubacterium rectale[2]. Lowered gut microbiota diversity has been observed in association with metabolic aberrations, including insulin resistance and dyslipidemia. Furthermore, several risk factors, such as hypertension, diabetes mellitus and hyperlipidemia, can lead to stroke. It is because of the influence of pretty confounding variable of cardiovascular risk factors among groups that we included the healthy controls with hypertension, diabetes mellitus or hyperlipidemia to exclude the influence of specific comorbidities and the baseline characteristics of the three groups were consistent and comparable (Table 1; all P > 0.05). We were very sorry that our inappropriate description caused your confusion.

The revised text reads as follows on Page 6, line 138:

The HC group was defined as volunteers absent from parenchymal lesions of the major organs and past medical history of cardio-cerebrovascular disease through physical examination.

The baseline characteristics of the three groups were consistent and comparable, including age, sex, the body mass index, and comorbidities in order to exclude the influence of specific comorbidities (hypertension, diabetes, and hyperlipidemia) and other confounders that affect gut microbiota (Table 1; all P > 0.05). 

Reference:

[1]Wang J., Ma Q., Li Y., et al. Research progress on traditional Chinese medicine syndromes of diabetes mellitus. Biomedicine & Pharmacotherapy. 2020;121:p. 109565. doi: 10.1016/j.biopha.2019.109565.

[2]Liu J, An N, Ma C, Li X, Zhang J, Zhu W, Zhang Y, Li J. Correlation analysis of intestinal flora with hypertension. Exp Ther Med. 2018 Sep;16(3):2325-2330. doi: 10.3892/etm.2018.6500.

[3] Le Chatelier E., Nielsen T., Qin J., Prifti E., Hildebrand F., Falony G., Almeida M., Arumugam M., Batto J.M., Kennedy S., et al. Richness of human gut microbiome correlates with metabolic markers. Nature. 2013;500:541–546. doi: 10.1038/nature12506.

[4]Bello M.G.D., Knight R., Gilbert J.A., Blaser M.J. Preserving microbial diversity. Science. 2018;5:33–34. doi: 10.1126/science.aau8816.

Q2.The inclusion and exclusion of various subgroup studies (tAIS, ntAIS etc.) have to been fully rationalized. What criteria were utilized to differentiate between severe or not severe cases? 

Author response: We appreciated your considerate suggestions very much. Phlegm-heat syndrome (PHS) consists of two syndrome elements, phlegm-damp and heat-flaming. We applied the Diagnostic Scale of Syndrome Elements in Ischemic Stroke (DSSEIS) to access the score of syndrome elements. The sensitivity and specificity of this scale has been confirmed through two rounds of multi-center clinical validation. Presently, the scale has been intensively used in clinical practice. The scale consists of 6 syndrome elements including inner wind, heat-flaming, phlegm-damp, blood stasis, qi-deficiency and yin-deficiency. The phlegm-damp and heat-flaming syndrome consists of 17 and 21 items, respectively. These items are essentially described about TCM symptoms, tongue coating and pulse condition. As long as two syndrome elements score of phlegm-damp and heat-flaming ≥10 could fulfill the diagnostic criteria of PHS. The inclusion criterion of the ntAIS group was to have a phlegm-damp and heat-flaming syndrome element score of ≤4, while the two syndrome element score of ≥10 was required for inclusion in the tAIS group. The scale was used to determine whether the syndrome elements were established but not able to judge the severity of the syndrome elements. As for the differentiation of severity of AIS patients, the NIHSS score indicates the degree of neurological function damage and is frequently used to determine the severity of the patient's condition on admission[1]. As such, we applied the NIHSS score to determine the severity of patients on admission in this study.

Reference:

[1]Wu Z, Zeng M, Li C, Qiu H, Feng H, Xu X, Zhang H, Wu J. Time-dependence of NIHSS in predicting functional outcome of patients with acute ischemic stroke treated with intravenous thrombolysis. Postgrad Med J. 2019;95:181–186. doi: 10.1136/postgradmedj-2019-136398.

Q3.The significance of the variations in microbial population in these diseased conditions need to be appropriately commented on.

Author response: We appreciated your valuable suggestions very much. We added the significance of the variations in microbial population into sections discussion to further elaborate the significance of the variations in microbial population. 

The revised text reads as follows on Page 17, line 455.

This was the first study to explore the characteristics of the gut microbiota in AIS patients with PHS. Ruminococcaceae_UCG_002 and Christensenellaceae_R-7_group could be applied to determine and differentiate the profile of PHS, was a key finding of this study and provided novel insight for a reasonable explanation on the cause of constipation and more severe neurological dysfunction in the AIS patients with PHS,which suggested that early diagnosis and early intervention was of significant implication in clinical practice. Additionally, this study also suggested that supplementation of the diet with fiber, butyrate, or probiotics containing butyrate-producing bacteria could exert the neuroprotective effect on stroke recovery via promotion of consolidate epithelial integrity. 

Reviewer 1 

Author response: Thank you so much for your careful comment and review, which helped to revise and improve the manuscript. We hope the manuscript, after careful revisions, meets your high standards. The authors welcome further comments if any. Below we provide the point-by-point responses.

Q1.  Line 37: Adding one line to explain the connection between AIS and PHS would be helpful.

Author response: Thank you so much for your advice, which we have now modified.

The revised text reads as follows on Page 2, line 24:

Phlegm-heat syndrome (PHS), a specific pathological state of the AIS, is one of the common traditional Chinese syndromes of stroke.

Q2. Line 76: affected

Author response: We appreciated your considerate suggestions very much. We have removed the spaces between the letters.

Q3. Line 100: controls description. Non AIS, Non PHS?

Author response: Thank you for your valuable advice. Considering the influence of pretty confounding variable of cardiovascular risk factors among groups, the healthy controls we included were relatively healthy with hypertension, diabetes mellitus or hyperlipidemia in order to exclude the influence of specific comorbidities. Briefly, the healthy controls were the populations with non AIS. We have added the definition of the healthy controls as you suggested.

The revised text reads as follows on Page 6, line 138:

The HC group was defined as volunteers absent from parenchymal lesions of the major organs and past medical history of cardio-cerebrovascular disease through physical examination.

Q4. Line 114: The absence of cardiovascular risk factors in the healthy control group makes them a pretty confounding variable compared to the other two groups, whereby conclusions about differences between HC and either of the other groups has to really be taken with a lot of salt. Perhaps it would be prudent to further elucidate why the decision was made to exclude non-stroke patients, who had hypertension/diabetes/hyperlipidemia.

Author response: Thank you so much for your valuable advice. We were sorry that our unclear description made you puzzled by the definition of healthy control group. The HC group in this study was relatively healthy. Recent studies have indicated that the changes of intestinal flora were closely associated with diabetes, hypertension and hyperlipidemia. Intestinal microbiota disorder, such as the decrease of beneficial bacteria and the increase of harmful bacteria, is easy to occur in diabetes patients[1].Additionally, there are abnormal changes in the content of intestinal floras in patients with hypertension including the decreased quantities of Bifidobacterium and Bacteroides thetaiotaomicron and the increased number of Eubacterium rectale[2]. Lowered gut microbiota diversity has been observed in association with metabolic aberrations, including insulin resistance and dyslipidemia. Furthermore, several risk factors, such as hypertension, diabetes mellitus and hyperlipidemia, can lead to stroke. It is because of the influence of pretty confounding variable of cardiovascular risk factors among groups that we included the healthy controls with hypertension, diabetes mellitus or hyperlipidemia to exclude the influence of specific comorbidities and the baseline characteristics of the three groups were consistent and comparable (Table 1; all P > 0.05). We were very sorry that our inappropriate description caused your confusion.

The revised text reads as follows on Page 6, line 138:

The HC group was defined as volunteers absent from parenchymal lesions of the major organs and past medical history of cardio-cerebrovascular disease through physical examination.

The baseline characteristics of the three groups were consistent and comparable, including age, sex, the body mass index, and comorbidities in order to exclude the influence of specific comorbidities (hypertension, diabetes, and hyperlipidemia) and other confounders that affect gut microbiota (Table 1; all P > 0.05). 

Reference:

[5]Wang J., Ma Q., Li Y., et al. Research progress on traditional Chinese medicine syndromes of diabetes mellitus. Biomedicine & Pharmacotherapy. 2020;121:p. 109565. doi: 10.1016/j.biopha.2019.109565.

[6]Liu J, An N, Ma C, Li X, Zhang J, Zhu W, Zhang Y, Li J. Correlation analysis of intestinal flora with hypertension. Exp Ther Med. 2018 Sep;16(3):2325-2330. doi: 10.3892/etm.2018.6500.

[7] Le Chatelier E., Nielsen T., Qin J., Prifti E., Hildebrand F., Falony G., Almeida M., Arumugam M., Batto J.M., Kennedy S., et al. Richness of human gut microbiome correlates with metabolic markers. Nature. 2013;500:541–546. doi: 10.1038/nature12506.

[8]Bello M.G.D., Knight R., Gilbert J.A., Blaser M.J. Preserving microbial diversity. Science. 2018;5:33–34. doi: 10.1126/science.aau8816.

Q5. Line 120: Is the recipe information included in the supplementary information?

Author response: Thanks for your considerate reminders. The registered nutritionist in Dongzhimen hospital provided the standardized recipe designed according to the Chinese Food Guide Pagoda for stroke patients. Diets formulated in this recipe could also meet the daily energy requirements for healthy people. Three meals per day for all the participants in this study were prepared strictly according to the standardized recipe.The detailed recipe was presented in the re-uploaded supplementary material.

Q6. Line 175: Where it shows the difference between the combined tntAIS and Chao1, does this not simply negate the value of including PHS in the analysis. Furthermore again we are confronted with he issue of confounds between chao1 and tntais. This is a serious flaw in the study design that needs to be addressed in terms of the logic behind combining the tAIS and ntAIS as well as the decision to exclude the confounds in the HC group.

Author response: Thank you for pointing this out. Honestly, we wanted to drawed the conclusion of this study based on two key comparisons. First, the comparison between tAIS and ntAIS group contributed to the influence of PHS on the alterations of alpha-diversity. Second, we combined the tAIS and ntAIS group into a single tntAIS group, which were the patients with acute ischemic stroke. By comparison between tntAIS and HC groups, we could found the influence of acute ischemic stroke on gut microbiota. Species richness could be estimated with the Chao1 and shannon estimator. As shown in table 2, the shannon index were higher in tAIS group than those in ntAIS group with statistically significant difference. Additionally, the chao1 index were significantly higher in tntAIS group(acute ischemic stroke patients) than those in HC group (Table 3). Therefore, it could be concluded that acute ischemic stroke may result in the increased species richness of gut microbita, especially in AIS patients with PHS. As for the influence of pretty confounding variable of cardiovascular risk factors among groups, we included the healthy controls with hypertension, diabetes mellitus or hyperlipidemia to exclude the influence of specific comorbidities and the baseline characteristics of the three groups were consistent and comparable (Table 1; all P > 0.05). In order to avoid the confusion, we modified the manuscript as you suggested.

The revised reads as follow on Page 12, line 292.

The shannon index were higher in tAIS group than those in ntAIS group with statistically significant difference. Additionally, the chao1 index were significantly higher in tntAIS group(patients with AIS) than those in HC group. Therefore, it could be concluded that AIS may result in the increased species richness of gut microbita, especially in AIS patients with PHS. The results of multiple diversities used to analyze the alpha and beta diversity of the community indicated that the alterations in richness and structure of gut microbiota existed in patients with AIS and more prevalent in those patients who also showed signs of PHS.

Q7. Line 234: The NIHSS is a likert scale, not a continuous variable so what were the cutoffs used to characterize severity? Or were they treated as a continuous variable? Please clarify!

Author response: Thank you for pointing this out. Actually, the National Institutes of Health Stroke Scale (NIHSS) is a 15 item impairment scale used to measure stroke severity[1]. In the current National Stroke Foundation guidelines, the NIHSS is recommended as a valid tool to assess stroke severity in emergency departments. The NIHSS includes the following domains: level of consciousness, eye movements, integrity of visual fields, facial movements, arm and leg muscle strength, sensation, coordination, language, speech and neglect. Each impairment is scored on an ordinal scale ranging from 0 to 2, 0 to 3, or 0 to 4. Item scores are summed to a total score ranging from 0 to 42 (the higher the score, the more severe the stroke)[2-3]. Additionally, NIHSS is the most commonly used efficacy outcome for quantification of neurologic deficits in stroke-related meta analyses and was calculated as continuous variable and expressed as mean and standard deviations. Acute minor stroke in CHANCE study[4] was defined by a score of 3 or less at the time of randomization on the NIHSS. Therefore, stroke severity at admission in our study was assessed using the NIHSS. 

Reference:

[1] Kwah L K, Diong J. National institutes of health stroke scale (NIHSS)[J]. Journal of physiotherapy, 2014.

[2]Lyden P, Lu M, Jackson C, NINDS tPA Stroke Trial Investigators.Underlying structure of the National Institutes of Health Stroke Scale: results of a factor analysis. Stroke 1999; 30: 2347–54.

[3] Lyden PD, Lu M, Levine SR, Brott TG, Broderick J, NINDS rtPA Stroke Study Group. A modiﬁed National Institutes of Health Stroke Scale for use in stroke clinical trials: preliminary reliability and validity. Stroke 2001; 32: 1310–17.

[4]Wang Y, Wang Y, Zhao X, et al. Clopidogrel with aspirin in acute minor stroke or transient ischemic attack[J]. N Engl J Med, 2013, 369: 11-19.

Reviewer 2: 

Author response: We feel great thanks for your professional review work on our article. As you are concerned, there are several problems that need to be addressed. According to your nice suggestions, we have made corrections to our previous draft, the detailed corrections are listed below.

Q1. I have no issue with the methodology and experimental approach of this research. I believe the techniques used such as 16S rRNA sequencing and bioinformatics approach are solid and the data presented confirms the conclusion of the study.I have one minor recommendation: if possible, mention if the PCR amplification was performed one, in duplicate or triplicate for each subject.

Author response: Thank you for pointing out this valuable issue. We amended the manuscript considering your comments.

The revised text reads as follows on Page 7, line 163:

The PCR amplification was performed in duplicate in a total reaction volume of 20 µL: H2O, 13.25 µL; 10× PCR ExTaq Buffer, 2.0 µL; DNA template (100 ng/mL), 0.5 µL; prime1 (10 mmol/L), 1.0 µL; prime2 (10 mmol/L), 1.0 µL; dNTP, 2.0 µL; and ExTaq (5 U/mL), 0.25 µL. 

Q2.The statistical analysis of the data is adequate. However, in one instance in the text, it is mention that the bacteria in the microbiome 'differed, but did not attain significance'. I think this needs to be rectified. If the statistic test doesn't show a significant difference, the text should not state that there is still a difference as this could lead to confusion.Recommendation: adjust the text at lines 194-195 to address this.

Author response: We appreciate your considerate suggestions very much. We have modified the sentences as you suggested.

The revised text reads as follows on Page 11, line 232:

The relative abundance of Firmicutes and Bacteroidetes in the ntAIS and tAIS group showed a trend of lower values than that of the HC group. By contrast, the relative abundance of Actinobacteria and Proteobacteria in the ntAIS and tAIS group showed a trend of higher values than that of the HC group. However, they failed to reach the level of statistical significance. 

Q3. The manuscript is written in acceptable English. There is no noticeable grammar/syntax/spelling mistakes. However, there are a few typos.

Line 276: typo in the reference citation

Line 297: typo in the reference citation

Author response: We were really sorry for our careless mistakes and have corrected all these errors.

Q5. In the text body, Figure 3A is mentioned before Figure 2B. The text would flow better if figures were in order. If possible, please adjust the text or figure order. If not, kindly explain.

Author response: We appreciated your considerate suggestions very much. We have modified the figure order as you suggested.

Q6.Figure 2: Mention in the legend how the abundance was calculated.

Author response: Thanks for your considerate suggestions very much. We have revised the figure lengend as you suggested.

The relative species abundances were calculated as percentages of the total species abundances(the number of tags measured in the sample divided by the total number of tags).

3) Figure 3: a) It would be pertinent to add the 'N' value for each group in the figure to avoid going back to the text. b) Describe 'relative abundance' in methods or in the legend. c) It is very hard to see what data is compared to what in the figure in the figure I was provided. I would recommend to adjust the figure to emphasize that better, or to improve the quality/resolution of the figure.

Author response: Thank you for pointing this out. We have revised the legend of figure 3 as you suggested. We are sorry that the figures that you have seen were of poor quality. As such, we have enhanced the resolution of all figures and uploaded them again.

4) Figure 4: Mention what a low or high LDA score means.

Author response: Thank you so much for your valuable advice. We have modified the legend of Figure 4 as you suggested.

The LDA score was obtained by LDA (linear regression analysis); the larger the LDA score, the greater the influence of species abundance on the difference effect.

5) Figure 2-3-4-5: I have some concerns about overall figure quality. Figures are blurry and hard to read. Overall quality or resolution of the figures needs to be improved, especially for the text found in the figure which is hard to read.

Author response: We apologized for not being very considerate and provided the figures of poor quality. As such, we have enhanced the resolution of all figures and uploaded them again.

---

## [Decision Letter · Decision Letter 1]

11 Oct 2022

Uncovering the characteristics of the gut microbiota in patients with acute ischemic stroke and phlegm-heat syndrome

PONE-D-22-09974R1

Dear Dr. Gao,

We’re pleased to inform you that your manuscript has been judged scientifically suitable for publication and will be formally accepted for publication once it meets all outstanding technical requirements.

Kind regards,

Vasu D. Appanna

Academic Editor

PLOS ONE

Additional Editor Comments (optional):

Reviewers' comments:

Reviewer's Responses to Questions

**Comments to the Author**

1. If the authors have adequately addressed your comments raised in a previous round of review and you feel that this manuscript is now acceptable for publication, you may indicate that here to bypass the “Comments to the Author” section, enter your conflict of interest statement in the “Confidential to Editor” section, and submit your "Accept" recommendation.

Reviewer #1: All comments have been addressed

2. Is the manuscript technically sound, and do the data support the conclusions?

Reviewer #1: Yes

3. Has the statistical analysis been performed appropriately and rigorously? 

Reviewer #1: Yes

4. Have the authors made all data underlying the findings in their manuscript fully available?

Reviewer #1: Yes

5. Is the manuscript presented in an intelligible fashion and written in standard English?

Reviewer #1: Yes

6. Review Comments to the Author

Reviewer #1: The changes made have greatly improved the quality of the article and I am content with the review of changes made! Thank you for your thorough responses!!!

7. PLOS authors have the option to publish the peer review history of their article (what does this mean?). If published, this will include your full peer review and any attached files.

Reviewer #1: **Yes: **Dr. Sean-Crispin Thomas

---

## [Editor Report · Acceptance letter]

25 Oct 2022

PONE-D-22-09974R1 

 Uncovering the characteristics of the gut microbiota in patients with acute ischemic stroke and phlegm-heat syndrome 

Dear Dr. Gao:

I'm pleased to inform you that your manuscript has been deemed suitable for publication in PLOS ONE. Congratulations! Your manuscript is now with our production department. 

Kind regards, 

on behalf of

Dr. Vasu D. Appanna 

Academic Editor

PLOS ONE